On the presence of Dipturus nidarosiensis (Storm, 1881) in the Central Mediterranean area

Carbonara Pierluigi carbonara@coispa.it 1
Cannas Rita 2
Donnaloia Marilena 1
Melis Riccardo 2
Porcu Cristina 2
Spedicato Maria Teresa 1
Zupa Walter 1
Follesa Maria Cristina 2
1 COISPA Tecnologia & Ricerca, Stazione Sperimentale per lo Studio delle Risorse del Mare , Bari , Italy
2 Department of Life and Environmental Sciences, University of Cagliari , Cagliari , Italy
Dos Santos Antonina
Electronic publication date: 2019 May 27
Publication date: 2019
Volume: 7
Electronic Location ID: e7009
Received 2018 Feb 7; Accepted 2019 Apr 24
Copyright: ©2019 Carbonara et al.
Copyright year: 2019
Copyright holder: Carbonara et al.
License: This is an open access article distributed under the terms of the Creative Commons Attribution License, which permits unrestricted use, distribution, reproduction and adaptation in any medium and for any purpose provided that it is properly attributed. For attribution, the original author(s), title, publication source (PeerJ) and either DOI or URL of the article must be cited.
License URL: https://creativecommons.org/licenses/by/4.0/

Keywords: DNA barcoding, Biodiversity skate, Central Mediterranean, Dipturus nidarosiensis

Funding: Italian Ministry of Agriculture (MiPAAF), European Commission EU Reg. 199/2008 FAO—AdriaMed Project This work was supported by Italian Ministry of Agriculture (MiPAAF), European Commission (EU Reg. 199/2008) and FAO - AdriaMed Project. The funders had no role in study design, data collection and analysis, decision to publish, or preparation of the manuscript.

==============================
The Norwegian skate Dipturus nidarosiensis (Storm, 1881) has only recently been recorded in the western Mediterranean Sea along the coast of southern Sardinia, off Algeria and the Alboran Sea. The present study confirmed the presence of the species in the Central Mediterranean Sea by identifying morphometric, morphological features and molecular markers. Biological sampling was conducted from 2010 to 2016 on eight specimens collected through commercial landings, offshore observer programmes and scientific surveys in Adriatic and Ionian waters at depths between 320 and 720 m. The total lengths of the specimens (juveniles and adults) ranged from 268 to 1,422 mm, and their body weights ranged from 44.5 to 12,540.0 g. They showed morphometric features that corresponded to those of Norwegian skates in the Northeast Atlantic and the Western Mediterranean. In previous analyses, molecular data were obtained by mitochondrial COI sequences. The haplotype network showed the occurrence of a common haplotype (Hap_1) shared by the individuals from areas in the North Atlantic, Sardinian, Algerian and Spanish Mediterranean Sea areas but not South Africa. The occurrence of individuals in different stages of life (i.e., juveniles, sub-adults and adults) and sexual development (immature and mature) suggested the presence of a species with a permanent reproductive allocation in the deep waters of the Mediterranean, which was exposed to a low level of fishing exploitation. Indeed, the deep depth distribution of the species could be the reason for the absence of information about this species in onshore or offshore fishery data collection programmes and scientific surveys.

Introduction

Species discrimination is a challenging issue mainly because it is based only on morphological diagnoses. Indeed, species classification errors are frequent because of several factors, such as phenotypic plasticity, the presence of cryptic taxa, morphological changes linked to life stage and the high level of expertise required for this kind of analysis (Hebert et al., 2003).

The misclassification of skates is common (e.g.,  Serena, 2005; Frodella et al., 2016; Iglesias, Toulhoat & Sellos, 2010). It has been estimated that approximately 50 known species are still only partially described or completely undescribed (Coulson et al., 2011). Consequently, in recent decades, DNA-based species discrimination has been applied (Hebert, Ratnasingham & deWaard, 2003) in an increasing number of cases (Spouge, 2016). The use of molecular markers, specifically the cytochrome c oxidase I gene (COI), has proven to be very helpful on several occasions in the identification of ray and skate species in the Mediterranean, Atlantic and other waters worldwide (Ball et al., 2016; Bineesh et al., 2017; Cannas et al., 2010; Cariani et al., 2017; Cerutti-Pereyra et al., 2012; Coulson et al., 2011; Frodella et al., 2016; Griffiths et al., 2013; Iglesias, Toulhoat & Sellos, 2010; Landi et al., 2014; Lynghammar et al., 2014; Mabragana et al., 2011; Ordines et al., 2017; Ramírez-Amaro et al., 2017; Serra-Pereira et al., 2011; Spies et al., 2006; Vargas-Caro et al., 2016; Vella, Vella & Schembri, 2017; Ward et al., 2008; Ward et al., 2005; Martin Dia zDe Astarloa et al., 2008).

In this context, skates (Rajidae) display conservative morphological traits both among and between alive and fossil species (Valsecchi et al., 2005). Hence, for this particular taxonomic group, DNA barcoding has been used to describe new species (e.g., Last & Séret, 2016), extirpation (e.g., Ball et al., 2016) and species’ expansion range (e.g., Ordines et al., 2017). In particular, the misclassification problem is particularly crucial in the genus Dipturus (Griffiths et al., 2010; Iglesias, Toulhoat & Sellos, 2010). In the eastern Atlantic Ocean and in the Mediterranean Basin, such misclassification includes Dipturus batis (Linnaeus, 1758), Dipturus oxyrinchus (Linnaeus, 1758) and Dipturus nidarosiensis (Storm, 1881). There has been a high percentage of misclassification in the fishery data reported for the French coast (Iglesias, Toulhoat & Sellos, 2010), mostly among the following species: D. oxyrinchus, D. batis, D. nidarosiensis and Rostroraja alba (Lacepede, 1803). These findings could have led to the underestimation of the risk of a dramatic decline and collapse of the spawning stock (Griffiths et al., 2010; Iglesias, Toulhoat & Sellos, 2010). Hence, the risk of extinction of these species might be higher than previously calculated (Iglesias, Toulhoat & Sellos, 2010).

D. nidarosiensis is one of the species that is often misclassified in both the Atlantic and the Mediterranean Basin (Cannas et al., 2010; Follesa et al., 2011; Cariani et al., 2017; Ramírez-Amaro et al., 2017). The geographical distribution of the Norwegian skate D. nidarosiensis is known to occur along the northeast Atlantic, from Iceland, the Faroe Islands and northern Norway to Madeira and northern Mauritania (Stehmann, 1990). This species is considered endemic in the northeast Atlantic and the Bay of Biscay (Priede et al., 2010; Stehmann & Bürkel, 1984; Rodríguez-Cabello, Pérez & Sánchez, 2013). However, recently, it has been recorded in the western Mediterranean Sea, along the southern Sardinia coast (Cannas et al., 2010; Follesa et al., 2012; Cariani et al., 2017; Marongiu et al., 2017; Porcu et al., 2017), off Algeria (Cariani et al., 2017), in the Alboran Sea (Ramírez-Amaro et al., 2017) and in the southern Adriatic and Ionian Seas (Cariani et al., 2017). The first catch of Norwegian skate specimens in western Mediterranean waters (2005) was reported to have occurred in Sardinia (Cannas et al., 2010); however, the records of the Alboran Sea are more recent (2012–2016; Ramírez-Amaro et al., 2017). The abovementioned authors hypothesised the ancient presence of the Norwegian skate, which had not been detected due to a misclassification between D. nidarosiensis and D. batis (Cannas et al., 2010).

In the IUCN Red List, the Norwegian skate D. nidarosiensis is listed as near-threatened (Stehmann et al., 2015); moreover, this species is rare in both the northeast Atlantic and the Mediterranean areas (Williams, Helle & Aschan, 2008; Cannas et al., 2010). Moreover, because the Norwegian skate has biological characteristics that are similar to other elasmobranchs, such as low fecundity, slow growth and high length at first maturity (Follesa et al., 2012), it is particularly vulnerable to environmental and anthropogenic pressures. Until now, information about the life history of the Norwegian skate in the Mediterranean Basin has been limited and based only on specimens caught around Sardinia Island (i.e., the central-western Mediterranean) (Follesa et al., 2012).

The present study is the first, from morphological and genetic points of view, to describe and characterise the Norwegian skate specimens caught in the southern Adriatic Sea and the north-western Ionian Sea (Central Mediterranean Basin), where this species has never been reported (Marano et al., 2002; Sion et al., 2003; Sion et al., 2004; Zupa et al., 2010; Relini et al., 2000; Relini et al., 2010) until very recently (Cariani et al., 2017). This study provides further information about the geographical distribution and the life-history traits of this species. Morphological comparisons with specimens from Sardinian waters and the Alboran Sea are made to provide the correct classification of this species in the Mediterranean Basin (Cannas et al., 2010; Follesa et al., 2012; Ramírez-Amaro et al., 2017).

Methods

Sampling

Samples of D. nidarosiensis were collected on land (e.g., market sampling) along the ports of the Apulia region and, from offshore observers (i.e., discard monitoring) in a fishery sampling programme (Data Collection Framework [DCF]; EU Reg. 1543/2000, 1639/2001, 1581/2004, 199/2008 and 1004/2017) and from fisheries independent scientific surveys (Mediterranean International Trawl Survey - MEDITS) (AAVV, 2017) (Fig. 1). In total, eight specimens were morphologically analysed and preserved in the deep freezer (−20 °C) at the COISPA Institute Laboratory. Morphology and colour analysis as well as morphometric measurements (nearest 1 mm) (Fig. 2), weights (the nearest 1 g) and meristic counts were performed following the previous descriptions of skate species (Ebert & Stehmann, 2013; Serena, 2005). All measurements and meristic counts are shown in Fig. 2 and Table 1.

Figure 1 The geographical locations of the D. nidarosiensis catches.

Spatial distribution of Norwegian skate Dipturus nidarosiensis caught during 2010–2016. The numbers correspond to the specimen codes reported in Table 1.

Figure 2 The morphometric measurements recorded for D. nidarosiensis specimens.

Schematic representation of the morphometric measurements recorded for Dipturus nidarosiensis specimens caught in the central Mediterranean Sea. Dorsal view at left (A) and ventral view at right (B). Total Length (TL); Disc Length (DL); Pre 1° dorsal fin Length (L1P); Pre 2° dorsal fin Length (L2P); Preorbital or snout length (Sporb); Preoral Length (SP); Interorbital Length (Dorb); Internostril Length (Dnos), Disc Width (DW); Tail Length (Ltail).

Sex and maturity were determined macroscopically, and the gonad weights were recorded to the nearest 0.1 g. Gonadal maturity was classified using the MEDITS maturity scale (AAVV, 2017; Follesa & Carbonara, 2019). The following results are shown in Table 2: Stage 1 (virgin immature); Stage 2 (maturing); Stage 3A (mature); Stage 3B (mature/extruding active); Stage 4A (resting); Stage 4B (regenerating).

Genetic analyses

Eight specimens were used in the genetic analysis, four of which were previously analysed by Cariani et al. (2017) (codes 1–4; Table 1) while the remaining four specimens (codes 5–8; Table 1) were analysed for the first time here.

The total genomic DNA was extracted from tissue samples using the salting-out protocol (Miller, Dykes & Polesky, 1988). These samples (skin and white muscle) were collected from the ventral side of the pectoral fin (ca. 0.5 cm3 in size). The tissue samples were preserved in 80% ethanol at 4 °C.

Table 1 The morphometrics, meristics, sexes, maturity, depth, dates of capture and origins data are reported.

Summary of data collected from commercial landings and scientific surveys, NA, not available.

Code specimens	1	2	3	4	5	6	7	8	
Date	29∕09∕2010	31∕07∕2011	31∕07∕2011	06∕10∕2011	13∕08∕2014	13∕08∕2014	02∕08∕2015	01∕08∕2016	
Source	Landing (Market)	Landing (Discards)	Landing (Discards)	Landing (Market)	MEDITS	MEDITS	MEDITS	MEDITS	
Depth (m)	610	600	600	720	377	550	373	320	
Morphometric measurements (mm)–Percentage of total length (%)	
Total Length (TL)	1,235	705	912	1194	269	,1422	356	305	
Width Disc (WD)	855–69.2	511–72.5	653–71.6	884–74.0	186–69.1	1,003–70.5	228–64.0	215–70.5	
Length Disc (LD)	720–58.3	415–58.9	524–57.5	713–59.7	146–54.3	842–59.2	187–52.5	172–56.4	
Length 1° dorsal fin (L1P)	1,020–82.6	560–79.3	723–79.3	977–81.8	206–76.6	1162–81.7	252–70.8	237–77.7	
Length 2° dorsal fin (L2P)	1,087–88	622–88.2	802 87.9	1049–87.9	229–85.1	1,259–88.5	282–79.2	266–87.2	
Distance interorbital (Dorb)	63–5.1	34–4.8	46–5	59–4.9	12–4.5	75–5.3	15–4.2	14–4.6	
Distance internostril (Dnos)	99–8	55–7.8	74–8.1	98–8.2	19–7.1	114–8	25–7	20–6.6	
Snout preorbital Sporb (Sporb)	246–19.9	141–20	181–19.8	242–20.3	51–18.9	286–20.1	67–18.8	63–20.7	
Snout preoral (SP)	228–18.5	133–18.9	173–19	215–18	56–20.8	314–22.1	69–19.4	68–22.3	
Length Tail (Ltail)	416–33.7	240–34	309–33.9	412–34.5	136–50.6	568–39.9	160–44.9	157–51.5	
Meristic counts (N ∘)	
Thorns on median row disc	20	12	12	15	0	19	0	0	
Median thorns on tail	32	18	23	33	9	35	10	11	
Lateral torn on tail	35	30	34	37	0	33	0	0	
Rows of teeth upper jaw	44	45	43	44	44	45	42	42	
Rows of teeth lower jaw	42	43	41	42	42	43	39	39	
Gill slits	5	5	5	5	5	5	5	5	
Weight (g)	
Total Weight	10,043	1,138	2,596	9,200	45	12,540	105	75	
Gonad weight	60	1.9	6.9	400	NA	58	NA	NA	
Liver weight	560	44.4	133.6	495.7	1.4	62.7	1.8	1.6	
Gutted weight	8,140	1,023.1	2,250.4	8,580	39.6	11,090	71.8	62.2	
Sex and Maturity	
Sex	Male	Male	Male	Female	Female	Female	Male	Female	
Maturity	3B	3A	3A	3A	1	2	1	1	

The primers for the amplifications of the mitochondrial COI gene (COI-3 primer cocktail) were obtained from Ivanova et al. (2007). The PCR reactions were set up in a 25-µl reaction volume containing 2.5 µl of 10X Taq Buffer (Thermo Scientific), 2.5 µl of 2 mM dNTPs, 2.5 µl of 25 mM of MgCl2, 0.1 µl of each (forward and reverse) 20 mM primers, 0.16 µl of Taq polymerase (Dream Taq® Thermo Scientific) and 1 µl of DNA (50–100 ng). The amplification was based on the following cyclic parameters: 5 min at 94 °C for the initial denaturation, followed by 40 cycles of 30 s at 94 °C, 40 s at 53 °C for the annealing of primers, 50 s at 72 °C for extension and then 7 min at 72 °C for the final extension. The PCR products were outsourced to Macrogen Europe (Netherlands) for sequencing. The sequences were obtained in both directions and submitted to GenBank under accession number MG699904–MG699907. The sequences were edited and aligned in MEGA v. 6 (Tamura et al., 2013) and then translated into aminoacidic sequences using the vertebrate genetic code to exclude the occurrence of codon stop and nuclear pseudogenes.

The number of haplotypes, haplotype diversity [hd], average number of nucleotide differences [k] and nucleotide diversity [π] were retrieved using DnaSP v. 5.1 (Librado & Rozas, 2009). Graphically, the haplotypes were arranged in a network with PopART (http://popart.otago.ac.nz) using the median-joining method (Bandelt, Forster & Röhl, 1999).

The sequences obtained in this study were compared to published COI sequences for the species from Mediterranean and eastern Atlantic Ocean individuals of the genus Dipturus (D. nidarosiensis, D. oxyrinchus, D. batis, D. cf flossada, D. cf intermedia), which are available in GenBank (https://www.ncbi.nlm.nih.gov/genbank) and BOLD (http://www.boldsystems.org/). Rostroraja alba was used as the outgroup. Table S1 provides the list and detailed information about the sequences used in the analyses. At least three sequences per species, such as D. intermediate and D. flossada, were used to a maximum of 42 sequences for D. oxyrinchus.

Table 2 The description of the maturity scale, used in this study.

Maturity scale for the Elasmobranch oviparous species (AAVV, 2017; Follesa & Carbonara, 2019).

Stage	Maturation state	Reproductive apparatus aspect	
		Females	Males	
1	IMMATURE/VIRGIN	Ovaries barely discernible with small isodiametric follicles. Distal part of oviducts thick-walled and whitish. Oviducal glands not clearly visible.	Small and flaccid claspers not reaching the posterior edge of the pelvic fins. Spermducts not differentiated. Small and narrow testes.	
2	MATURING	Whitish and/or few yellow maturing follicles visible in the ovary. Distal part of oviducts (uteri) well developed but empty. Developing oviducal glands.	Clasper extending past the posterior edge of the pelvic fins. Spermducts well developed, and starting to meander.	
3A	MATURE	Ovaries with yellow follicles (large yolk follicles). Enlarged oviducal glands and distended oviducts.	Hard and ossified claspers extendings well beyond the posterior edge of the pelvic fin. with internal structure is generally. Testes greatly enlarged. Spermducts meandering over almost their entire length.	
3B	MATURE/EXTRUDING/ACTIVE	Ovaries walls transparent. White or yellow follicles. Large oviducal glands. Egg-cases more or less formed in the oviducts (Extruding Stage).	Clasper longer than tips of posterior pelvic fin lobes, skeleton hardened with axial cartilages hardened and pointed. Spermducts largely. Sperm flowing on pressure from cloaca (Active Stage).	
4A	RESTING	Ovaries walls transparent. White or yellow follicles. Enlarged oviducts, collapsed and empty. Reduced oviducal glands diameter.	Clasper longer than tips of posterior pelvic fin lobes, skeleton hardened withaxial cartilages still hardened. Spermducts empty and flaccid.	
4B	REGENERATING	Ovaries full of small follicles similar to stage 2, enlarged oviducal glands and uteri	Clasper longer than tips of posterior pelvic fin lobes, skeleton hardened withaxial cartilages still hardened. Spermducts empty and flaccid.	

The relationships among haplotypes were investigated using two approaches: the Bayesian method using MrBayes v. 3.1 (Huelsenbeck & Ronquist, 2001; Ronquist & Huelsenbeck, 2003) and the neighbour-joining method in MEGA using Kimura 2-parameter genetic distances (K2P; Kimura, 1980) and a bootstrap test with 1,000 replicates. In MrBayes, the analyses were performed using two parallel runs of two million generations each, using four chains, sampling every 100 generations, a burn-in of 0.25, and saving branch lengths. The performance of the analyses was evaluated using Tracer v. 1.6 (Rambaut et al., 2014). The trees were visualised by MEGA.

Ethical compliance

All the Norwegian skate (Dipturus nidarosiensis) analysed in this study, which were drawn from commercial fishing activity (DCF; EU Reg. 1004/2017), did not fit the European Commission recommendations (Directive 2010/63/EU of the European Parliament and of the Council of 22 September 2010) or Italian National Law (Decree Law n. 26 of 4 March 2014) regarding the protection of animals used for scientific experiments. Moreover, according to the welfare guidelines under the Decree Law n. 26 of 4 March 2014 (Italian Ministry of Health) and the protocol approved by the Committee on the Ethics of Animal Experiments of COISPA (Italian Ministry of Health 15/2015-UT), the specimens brought alive onboard during the scientific survey or the monitoring on the commercial fishing vessel should be killed using an overdose of anaesthetic. However, all the animals sampled in this study were already dead at the time of sampling: both the specimens sampled at the landing and on board (i.e., discard monitoring and scientific survey when the specimens were brought on board).

Figure 3 Juvenile and adult specimens (dorsal-ventral view) of D. nidarosiensis.

Juvenile ((A): dorsal view; (B): ventral view) and adult female ((C): dorsal view; D: ventral view) of Dipturus nidarosiensis. Photo: Pierluigi Carbonara.

Results

The Norwegian skate specimens were captured around the pit of Bari and in the northwest part of the Ionian Sea (Fig. 2) at depths ranging between 320 and 720 m. All measurements, meristic counts, sex and maturity are presented in Table 1. All specimens showed uniformly dark ventral sides, including the ampullary pores (Fig. 3). The disc was wider (WD 64.0–74.0% of TL) than it was long (LD 52.5–59.7% of TL), and it was covered uniformly by a darkish mucous on both sides (Fig. 3). The teeth were pointed in parallel rows in the males and in molar-like sets in pavement in the females (Fig. 4), with the numbers of tooth rows on the upper and lower jaws between 42–45 and 39–43, respectively. It appeared that the number of thorns on the tail increased with the length of the specimens, as shown in Table 1. Moreover, both the dorsal and ventral sides were prickly in the area of the snout (Fig. 5). Morphometric variability between the juveniles and adults was detected in the relationships between the Length Tail (Ltail)/Total Length and Distance interorbital (Dorb)/Total Length (Table 1). In terms of the percentage of TL, Ltail was longer in the juveniles at about half of the TL, and it was one-third of the TL in the sub-adults and adults (Fig. 2). On the contrary, the Dorb increased with the increasing of TL. Moreover, the juveniles did not show spines on the median line of the disc or on both lateral sides of the tail, which were present in the adults (Table 1).

Figure 4 The mouth of female and male D. nidarosiensis specimens.

Detail of the mouth of (A) female with molar-like teeth and (B) male with pointed teeth. Photo: Pierluigi Carbonara.

Figure 5 The snout of D. nidarosiensis.

Detail of the snout (A) dorsal and (B) ventral side. Photo: Pierluigi Carbonara.

The juveniles (maturity stage 1) were caught in shallow waters at depths between 320 and 377 m, whereas the sub-adult (maturity stage 2) and adult specimens (maturity stage 3A and 3B) were caught in depths ranging between 550 and 720 m.

Furthermore, all samples caught during the MEDITS trawl survey were associated with a biocenosis typical of compact bathyal muds. In particular, the juveniles (specimens 5, 7 and 8; Table 1) caught at upper depths were found to be in association with a biocenosis characterised by tall sea pen Funiculina quadrangularis facies.

Genetic analyses

A 633 bp fragment of the COI gene was obtained from eight individuals from the South Adriatic and Ionian seas. The combination of these data with those available for D. nidarosiensis in other areas in the Mediterranean and the adjacent Atlantic Ocean provided the final alignment of 38 sequences and seven haplotypes (555 bp; Hd: 0.515; k: 0.586; π: 0.00106). The haplotype network (Fig. 6) showed the occurrence of a common haplotype (Hap_1) shared by 26 individuals from all investigated areas except South Africa. It also showed six additional haplotypes, four of which were exclusive to the Mediterranean (Hap_2-Hap_5), and two were exclusive to the Atlantic (Hap_6-Hap_7).

Figure 6 Median-Joining network of the COI haplotypes.

Each circle represents a haplotype, and the area of the circle is proportional to the haplotype frequency. All mutational steps are equal to 1 and are represented with a vertical line. Colours correspond to different areas of origin of the sequence (SAF, South Africa; ATL, Northeast Atlantic; ALB, Alboran Sea; ALG, Algeria; SAR, Sardinia; IAD, Ionian + Adriatic seas). Haplotype code correspondence is shown in Table S1.

The phylogenetic analyses (Fig. 7) further confirmed the identification of the Ionian–Adriatic specimens as D. nidarosiensis. They were clustered in a strongly supported clade that was clearly distinguishable from the other species of Dipturus found in the Mediterranean and the eastern Atlantic Ocean. The sequences that were the most similar to D. nidarosiensis were those of D. cf flossada (95% base pair similarity), and the least similar were those of R. alba (87.82% base pair similarity).

Figure 7 Phylogenetic tree.

Phylogenetic tree based on mitochondrial COI sequences. Bootstrap/posterior probability values >70 are shown next to the nodes for the Neighbour Joining and Bayesian methods, respectively. In red the clade were the Ionian-Adriatic sequences clustered.

Discussion

The results of the present study confirmed the presence of the Norwegian skate Dipturus nidarosiensis in the central Mediterranean Sea by identifying morphometric and morphological features and molecular markers. The morphological parameters described are in line with the data collected in Sardinian waters (Cannas et al., 2010) and the Alboran Sea (Ramírez-Amaro et al., 2017). In particular, the relationship between the morphometric measurements and total length was similar to the data reported in the Alboran Sea (Ramírez-Amaro et al., 2017). Specifically, the measurements of the juveniles from the Spanish area (Ramírez-Amaro et al., 2017) were in accordance with those of the Adriatic Basin. Furthermore, the high variability in Length Tail during the different stages of life seemed to be common to other Rajidae species, including R. clavata, D. oxyrinchus and Amblyraja radiata (Mnasri et al., 2009; Capapé et al., 2015; Templeman, 1987).

The genetic analysis of the mitochondrial gene COI made it possible to confirm the morphological identification of the Adriatic–Ionian samples as D. nidarosiensis. In particular, in this study, the molecular data revealed that the Mediterranean specimens of D. nidarosiensis (from Sardinia, Algeria and the Alboran Sea) shared COI mitochondrial haplotypes with the Atlantic specimens (see Table S1 for details). However, additional analyses based on more powerful markers (i.e., microsatellites and SNPs) and larger numbers of individuals are needed to investigate possible intraspecific differences and genetic connectivity among the different areas within the Mediterranean as well as between the latter and the Atlantic populations (Griffiths et al., 2011).

The D. nidarosiensis reported in this paper were captured mainly around the Bari Pit (maximum depth = 1,233 m) in an area between the western Italian coast and the eastern Montenegrin and Albanian coasts. This area is connected with the northern part of the Ionian Sea through the Otranto Channel, and it is part of a system of currents that exchange and mix deep water from the eastern part of the Mediterranean Sea and the northern part of the Adriatic Sea (Vilibic & Orlic, 2002). Although the capture locations of Norwegian skates are dispersed over 200 miles, all these areas are connected by deep-water currents (Artegiani et al., 1997). In the southern Adriatic and Ionian bathyal systems, there is an absence of barriers to gene flow in deep species as well as Aristeus antennatus (Marra et al., 2015). In the case of A. antennatus, it was previously hypothesised that the presence of the population could be structured into two layers: a deeper stock not affected by fishing, which preserves genetic homogeneity; and an upper stock living in the fishing ground. The larval dispersal, adult migration and continuous movements of individuals from deeper grounds to upper areas could explain their genetic homogeneity throughout the Adriatic and Ionian system (Marra et al., 2015). Similarly, the partial overlap between the Norwegian skate’s habitat and fishing grounds in the Adriatic and Ionian Seas suggests that the Adriatic and Ionian D. nidarosiensis specimens are part of the same population.

The occurrence of the Norwegian skate in the southern Adriatic Sea and northern Ionian Sea sheds new light on the dispersion of this species in the Mediterranean Basin. In interviews with fishermen in the southern Adriatic, it emerged that the occurrence of this species, locally called “black monk”, is not unusual.

However, the presence of this species in the Mediterranean was unnoticed until 2010 (Cannas et al., 2010), which could be attributed to two main reasons. Firstly, it had been misclassified with others species of the Dipturus genus (Cannas et al., 2010) as well as for the French coast (Iglesias, Toulhoat & Sellos, 2010). Indeed, the presence of the Dipturus species is not recent (Griffiths et al., 2011). In the southern Adriatic, catches of D. oxyrinchus were reported in 1994–1995 (Relini et al., 2000; Marano et al., 2002), and in the western Ionian, catches of D. batis and D. oxyrinchus were recorded in 1992 and 2000–2001, respectively (Relini et al., 2000; Sion et al., 2003). In the central-western part of the basin, D. oxyrinchus has been caught since the first surveys that were funded by the DCF in mid-1980 (Relini et al., 2000; Relini et al., 2010). Secondly, in the southern Adriatic, the bottom trawl fleet fishing grounds are concentrated on the shelf and the upper part of slope (Mannini & Sabatella, 2015). In the western Ionian, as in the other part of the basin, they are located mainly on the slope to a depth of 800 m (Russo et al., 2017). Hence, the catches of D. nidarosiensis characterised by a wider depth distribution (between 300 and 1,700 m) (Cannas et al., 2010; Follesa et al., 2012; Ramírez-Amaro et al., 2017) could be unrecorded. In the studied areas, the depth of the fishing activity only partially overlaps the deep distribution of the Norwegian skate. In addition, in the Mediterranean sites where D. nidarosiensis specimens were found, the seabed presents similar characteristics (e.g., canyons and slopes of high depths), which are favourable to this species but unfavourable to the trawling activity of both commercial (Mannini & Sabatella, 2015) and research vessels (AAVV, 2017).

The presence of maturing and mature males and females in early autumn (end of September/beginning of October) and summer, respectively, could support previous preliminary observations on the reproductive biology of the species in Sardinian waters (Follesa et al., 2012). Furthermore, the collection of empty egg cases on the sea floor as well as individuals in the different life stages (embryos within egg case, juveniles, sub-adults and adults) of both sexes suggests the presence of a species with a permanent reproductive allocation in the Mediterranean Sea (Cannas et al., 2010; Follesa et al., 2012; Cariani et al., 2017; Porcu et al., 2017; Ramírez-Amaro et al., 2017).

Based on the data recorded, despite the few specimens analysed, the Norwegian skate showed that a bathymetric segregation of juveniles occurred only in the upper part of the slope, and sub-adults and adults occurred at lower depths, which was previously reported in Sardinian waters (Follesa et al., 2012). This deep segregation by the ontogenic phase was also observed in other species of skates, including Raja clavata, R. asterias and Rhinoptera bonasus (Rousset, 1990; Ferrá et al., 2016; Fisher, Call & Grubbs, 2013; Ellis et al., 2005). In addition, the finding of Ionian and Adriatic D. nidarosiensis specimens on compact mud bottoms confirmed that the species prefers this type of sea bed, which was also observed by Follesa et al. (2012) in Sardinian waters. Moreover, the present study is the first to record the presence of juveniles in association with a biocenosis characterised by bathyal muds with tall sea pen Funiculina quadrangularis facies.

Nevertheless, several aspects of D. nidarosiensis remain poorly investigated. For this reason, it would be useful to continue the monitoring and sampling of this species in order to collect additional data to clarify aspects of its biology (e.g., reproduction and growth) and ecology (e.g., trophic relationship and geographical distribution).

Conclusion

The results obtained in this study confirm the presence of Dipturus nidarosiensis in the central Mediterranean Sea (southern Adriatic and northwest Ionian seas) and provide information about its geographical distribution, life-history and morphological traits in the Mediterranean Basin. Indeed, misclassification of the Mediterranean skate species has occurred frequently (Iglesias, Toulhoat & Sellos, 2010), resulting in the misinterpretation of several biological characteristics. The morphological data were similar to those recorded in Sardinia and Alboran Sea (Cannas et al., 2010; Ramírez-Amaro et al., 2017). Moreover, the morphological differences between juveniles and adults were highlighted in terms of morphological and meristic characteristics. The genetic analysis conducted through the mitochondrial gene COI (Spouge, 2016) made it possible to confirm the morphological identification of the Adriatic–Ionian samples as D. nidarosiensis.

Supplemental Information

Table S1 Raw data related to the genetic analysis (Species, GenBank accession numbers, Haplotype code, Country, Origin, Reference)

Click here for additional data file.

Supplemental Information 2 DNA sequences

COI DNA nucleotide sequences of Dipturus nidarosiensis newly obtained in the present study (GenBank accession numbers MG699904 –MG699907).

Click here for additional data file.

The authors are grateful to the Academic Editor Dr. Antonina Dos Santos and three anonymous reviewers for their constructive comments and suggestions, which greatly helped to improve the manuscript.

Additional Information and Declarations

Competing Interests

Author Contributions

Animal Ethics

DNA Deposition

Data Availability

The authors declare there are no competing interests.

Pierluigi Carbonara conceived and designed the experiments, performed the experiments, analyzed the data, prepared figures and/or tables, authored or reviewed drafts of the paper, approved the final draft.

Rita Cannas conceived and designed the experiments, performed the experiments, analyzed the data, contributed reagents/materials/analysis tools, prepared figures and/or tables, authored or reviewed drafts of the paper, approved the final draft.

Marilena Donnaloia performed the experiments, authored or reviewed drafts of the paper, approved the final draft.

Riccardo Melis analyzed the data, contributed reagents/materials/analysis tools, prepared figures and/or tables, authored or reviewed drafts of the paper, approved the final draft.

Cristina Porcu analyzed the data, contributed reagents/materials/analysis tools, authored or reviewed drafts of the paper, approved the final draft.

Maria Teresa Spedicato conceived and designed the experiments, authored or reviewed drafts of the paper, approved the final draft.

Walter Zupa performed the experiments, analyzed the data, prepared figures and/or tables, authored or reviewed drafts of the paper, approved the final draft.

Maria Cristina Follesa conceived and designed the experiments, analyzed the data, contributed reagents/materials/analysis tools, prepared figures and/or tables, authored or reviewed drafts of the paper, approved the final draft.

The following information was supplied relating to ethical approvals (i.e., approving body and any reference numbers):

All Norwegian skate (Dipturus nidarosiensis) analysed in this study from commercial fishing activity (DCF; EU Reg. 1004/2017) do not fit under the European Commission recommendations (Directive 2010/63/EU of the European Parliament and of the Council of 22 September 2010) or Italian National Law (Decree Law n. 26 of 4 March 2014) regarding the protection of animals used for scientific experiments. Moreover following the welfare guidelines under the Decree Law n. 26 of 4 March 2014 (Italian Ministry of Health) and the protocol approved by the Committee on the Ethics of Animal Experiments of COISPA (Italian Ministry of Health 15/2015-UT), the specimens brought on-board alive, during the scientific survey or the monitoring on the commercial, should be killed with an overdose of anaesthetic. However all the animals sampled in this study were already dead at the time of sampling both the specimens sampled at the landing and on board (discard monitoring and scientific survey when the specimens brought on-board).

The following information was supplied regarding the deposition of DNA sequences:

Data is available at GenBank, accession numbers MG699904 to MG699907.

The following information was supplied regarding data availability:

Genetic analysis data (Species, GenBank accession numbers, Haplotype code, Country, Origin, Reference) are available in Table S1. Morphometrics data are available in the manuscript (Table 1).

The morphometrics data are included in Table 1.

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
