# Peer review of "On the presence of Dipturus nidarosiensis (Storm, 1881) in the Central Mediterranean area"

_PeerJ, doi:10.7717/peerj.7009_

## Round 0.1 · original submission · Major Revisions

I consider that the MS contains information with scientific value and that it deserves to be published. However, I agree with the reviewers opinion, that the work is poorly presented and that, especially the Introduction and discussion sections need a major revision. Please consider and address all comments made by the reviewers, especially, reviewers 1 and 3 which point to weakness in the MS that should be improved.

It is my intention to send the revised MS back to the reviewers.

Thank you for having considered PeerJ as an outlet for your work

Reviewer 1 ·

Basic reporting

Introduction
I feel that the introduction of the paper is poor and needs to be reorganized. I recommend the following points to develop:

a). DNA barcoding and its use for problematic species (See Hebert et al., 2003a,b).

b). Rajidae (skates) as a taxonomic problematic group due to their conservative morphological features (see Coulson et al., 2011). For this family, DNA barcoding has been used to the description of new species (e.g. Last & Séret, 2016), extirpation (e.g. Ball et al., 2016), as well as, expansion a species range (e.g. Ordines et al., 2017). Note that some of these examples are from lines 216-223.

c). Dipturus case. Add some examples about misidentification of this genus such as Dipturus batis (see Griffiths et al., 2010; Iglésias et al., 2010).

d). D. nidarosiensis misidentification. Development the study cases of this species in the Mediterranean (Cannas et al., 2010; Cariani et al., 2017; Ramírez-Amaro et al., 2017). Along these lines, the current distribution is the North East Atlantic Ocean (from Iceland, Faroe and northern Norway to Madeira and Mauritania) and the western Mediterranean Sea (from Alboran to southern Adriatic and Ionian seas). From lines 37-44.

e). Biology and red list of study species (enter information from lines 44-52).

f).The biology and ecology of D. nidarosiensis are poorly known in the Mediterranean, hence the importance of this paper.

Discussion
I feel that the discussion is poorly developed.
The main result of the present study is the expansion of geographic distribution of this species.
Remove lines 154-155.
Transfer lines 155-158 to the introduction
Lines 160-161: whose suggest that this species acceded through Gibraltar strait from the Atlantic Ocean?
Lines 161-164: according to the fishermen, the occurrence of this species in none usual, so it is likely this species has been confused with other Dipturus species. Develop this point.

In my view, the few and recent records of this species in the Mediterranean is due to the displacement of the trawling fleet toward deeper areas during the last years (see Marongiu et al., 2017). Develop this point.

Lines 176-177: add some examples from other species about the hypothesis that juveniles live in shallower water than adults.
Lines 188: add could be associated…
Lines 183-185: given the low number of samples analyzed, this paragraph is speculative.
Lines 201-202: long tail longer in juveniles than the adults apparently is common in skates, please add an example.
Lines 213-214: this hypothesis is speculative. Please provide some example of marine species where this connectivity pattern has been observed. These examples may reinforce their hypothesis.
Transfer lines 215-223 to the introduction.
Transfer lines 224-226-223 to the beginning of the discussion.

Proposed references:
• Hebert et al. (2003a) Proceedings of the Royal Society Series B Biological Sciences, 270, 313–321.
• Hebert et al. (2003b) Philosophical Transactions of the Royal Society of London Series B –Biological Sciences, 270, S96–S99.
• Coulson et al. (2011) Molecular Ecology Resources, 11, 968–978.
• Last and Séret (2016) Zootaxa, 4147, 477–489.
• Griffiths et al. (2010) Proceeding of the Royal Society Biological Science, 277, 1497–1503.
• Iglésias et al. (2010) Aquatic Conservation: Marine and Freshwater Ecosystems, 20, 319–333.

Experimental design

This manuscript proposes to use COI mitochondrial fragments to report D. nidarosiensis of Central Mediterranean. I am confused by the number of specimens analyzed. While in M&M are 8 specimens, then in results, the authors indicate that four of these samples were analyzed in a previous study (Cariani et al., 2017). It is necessary to clarify this point in M&M.
Lines 72: are the specimens deposited in a museum collection?
Lines 75-80: remove all these morphometric measurements and meristic counts from the text and only reference the table. Per example: All measurements are indicated in Table 1.
Lines 82-84: please describe, briefly, each maturity phase.
Lines 96: please add the software where the sequences were edited.
Lines 104-109: the authors used an adequate number of sequences to compare.

Validity of the findings

The authors argue that the results corroborate the hypothesis of the existence of the Norwegian in the Mediterranean, but this hypothesis has already been confirmed in previous studies (Cannas et al., 2010; Cariani et al., 2017; Ramírez-Amaro et al., 2017). So, this study implicates that this species extended its distribution to the Central Mediterranean.

The study fully provides the taxonomic and genetic identification of D. nidarosiensis in the area. All tables and figures are necessary and useful for readers. However, in the case of maturity phases, the authors could add pictures of internal reproductive traits and describe these in results. Genetic analyses (network and phylogenetic approaches) are well developed. Genetic identity percentage, according to the number of bases differences, could be added.

Additional comments

This article used morphological and genetic analyses to recorded, for the first time, Norwegian skate (Dipturus nidarosiensis) in the Southern Adriatic and Western Northern Ionian seas (Central Mediterranean basin). The authors argue that the results corroborate the hypothesis of the existence of the Norwegian in the Mediterranean, but this hypothesis has already been confirmed in previous studies (Cannas et al., 2010; Cariani et al., 2017; Ramírez-Amaro et al., 2017). So, this study implicates that this species extended its distribution to the Central Mediterranean.

Further, the authors claim that provide knowledge on its life history traits. I disagree with this objective. In my view, providing information on life history traits (e.g. fecundity, gestation period, age at maturity) implies a complex study, with more samples and applying statistical methods. In overall, I feel that the introduction and discussion of the paper are poor and needs to be reorganized. I hope they are of use to improve the clarity of the paper and the quality of the results and their interpretation.

Reviewer 2 ·

Basic reporting

The manuscript explores the presence of Dipturus nidarosiensis in the Central Mediterranean, reporting the first occurrences of the species in the Southern Adriatic Sea. A full morphometric description of the eight reported specimens and their genetic characterization and species identification are provided, making the manuscript self-contained.
The English language should be improved because phrase construction is sometimes very confusing (e.g., lines 53-57, 104-107, 158-163, 211-212), hence a full-text revision by a native-English speaker is strongly recommended.

Experimental design

The structure of the manuscript conforms to PeerJ standards.
The Introduction is appropriate, and literature is well referenced and relevant. The research objectives are clearly stated and were met.
Methods are well supported by figures, but the terminology used for the morphometric measurements is not accurate – please use Ebert and Stehmann (2013) for reference. Suggestion: Total length; Disk length; Pre-first dorsal-fin length; Pre-second dorsal-fin length; Preorbital or snout length; Preoral length; Interorbital length; Internostril length; Disk width; Tail length; (…).
Conclusions are well stated but could be summarised to avoid repeating the Discussion (e.g. Line 239, “Moreover, morphological differences between adults and juveniles were highlighted.”, deleting the rest of the sentence.)
Figures are relevant and have high quality, but their legends could be improved: include species and location so that the legends support the figures/table entirely without the need to consult the text – suggestion: “Figure 1. Schematic representation of the morphometric measurements recorded for Dipturus nidarosiensis specimens caught in the Central Mediterranean Sea.”
The following change in Table 1 editing is suggested: give Percentage of total length in brackets so that the values are not mistaken for a range.

Validity of the findings

This work is an original research and is within the scope of PeerJ.
The results bring important new insights on the distribution and biology of a poorly-recorded skate species in the Mediterranean Sea and the species identification is confirmed through genetic analyses.

Additional comments

I suggest the manuscript is accepted for publication but only after a careful English language revision. I opted not to make corrections in the sentence construction because I am not a native-English speaker either.

Please also take into consideration the following remarks:
1. Lines 18-19 and 39: The southern limit of distribution of the species is different between the Introduction (northern Mauritania) and the Abstract (northern Morocco) – please correct and include reference(s) in the Introduction.
2. Line 30: The correspondence between these life stages (which should further be ordered chronologically) and the maturity stages defined in lines 83-84 and in Table 1 is not clear.
3. Line 39: Correct “endemic to”.
4. Line 48: Correct “length at first maturity”.
5. Line 49: Complete the sentence: “(…) vulnerable to environmental and anthropogenic pressure” (for example).
6. Line 68: “landings and onboard sampling of commercial vessels (…) operating in the fishing ports (…)”
7. Lines 71-73: Re-order Figures 1 and 2.
8. Line 86: Clarify which tissues were used for DNA extraction.
9. Line 93: Correct “cycling” to “cyclic”.
10. Lines 103 and 112 and Figure 6: Write “Median-Joining” and “Neighbour-Joining” (hyphened).
11. Line 134 and Figure 4: Ebert and Stehmann (2013) describe teeth as follows: “Teeth in upper jaw in 41 to 44 rows, set in pavement in young and females, but in parallel rows in mature males.”
12. Lines 142-147: The haplotypes codes used in the text do not correspond to the codes used in Figure 6.
13. Lines 192-203: Mention that the values being analysed are in percentage of total length because it is erroneous to state that “Ltail is longer in juveniles than in adults” if absolute values are being considered.
14. Lines 192 and 237: Avoid using “comparable” because it might make the reader question why statistical analyses were not made. “Similar” is a suitable alternative.
15. Lines 204, 240 and Table 1: Correct “thorns”.
16. References: Invert order of references Marano et al. 2002 and Marongiu et al. 2017.
17. Figures 4 and 5: Correct “Particular” to “Detail”.

Reviewer 3 ·

Basic reporting

Please see general comments for the author.

Experimental design

Please see general comments for the author.

Validity of the findings

Please see general comments for the author.

Additional comments

Overall this manuscript is a worthwhile scientific contribution, however is not ready for publishing in its current form. It would need further work and suggest another round of reviewing.
It is strongly recommended that the authors undertake a major revision of the use of (Scientific) English throughout the whole manuscript and, having text reviewed prior to re-submission by a native English speaker would be beneficial.
The following examples are mere illustrations as the whole manuscript would need to be worked on and therefore, recommended a second review where more details can then be provided.
A more clear introduction would be preferable, with global species knowledge, problem specific to region, why this study would be beneficial including more knowledge about biology and the implications of reoccurrence in the Mediterranean have to any future management measures.

For example:
Line 26-29 for example something on the lines of “Juvenile and adult specimens were caught between 320 to 720 m depth, with total length ranging from 268 to 1422 mm and, total weight from 44.5 to 12540 g.”
Missing a stop at the end of sentence line 29.
Line 30 –“ the occurrence of individuals of both sexes and, at different life stages suggest ….”
Maybe avoid acronyms on the abstract, you could say from onshore and offshore programmes (I would imagine you mean market sampling data and observer programmes looking into discards data) and fisheries-independent scientific surveys.
Restructure sentence 27-29, suggest something like “Fish were caught around the pit of Bari in the Southern Adriatic Sea and, in the North West of the Ionian Sea.
First sentence of abstract similar to beginning of the Introduction, but in Introduction comes northern Mauritania and in the abstract Morocco. I would not necessarily start abstract with that sentence.
Throughout the entire manuscript et al is not in italics, it should be et al. ,
Line 39 – suggest something on the lines
Line 41 – references at the end of the sentence since some are repeated as they refer have data from different areas. Bit of consistency in naming convention e.g. “This species is considered endemic in the North East Atlantic and the Bay of Biscay” instead of North East Atlantic Ocean and Gulf of Biscay.
For example in line 46-47 “ Certaintly, in both the Atlantic and the Mediterranean” seems less formal than would be expected.
No consistency in the use of Atlantic (includes North West Atlantic, whole Atlantic), try to keep it to either Northeast Atlantic, in terms of Atlantic Coast all belong to NEA.
Methods section would need a more clear structure with English improved. “landings and embark on commercial vessels” could be something on the lines of “Samples of Dipturus nidarosiensis were collected through onshore (e.g. market sampling) and offshore observer programmes and, from fisheries-independent surveys.
Use colour instead of color.
Line 80 – not down jaw – “lower” jaw – not consistent with legend of photos
It would be beneficial to have maturity stages description, is this maturity widely used, how would you distinguish from stage 4a and stage 4b… not sure how it would be able to see regenerating on this species? Or even the resting… The stage seems more of a teleost species than off an elasmobranch. If we have the description we could also see what would be the difference between stage 3a and stage 3b. Recommendation to look into ICES Workshop 2012 were 4 stages were agreed for this type of species: A– Immature, B Maturing (though biologically still immature), C Mature, D Active.
The agreement within the ICES WGEF would be that these are still valid and should be the ones used.

Another example of the English would be in the Ethical compliance, would recommend to confirm if the journal guidelines for formatting laws would be acceptable. Also “in any case to minimize pains on animals, that arrived on the vessel” You could look into something as “However, specimens brought onboard alive during the scientific survey were subject of anaesthetic as to comply with fish welfare guidelines under the Decree Law etc….
Line 127 Ministry of Healt should be “Ministry of Health2

Results
Similar to other sections would benefit of restructure.
What would you classify in your study as juveniles, sub-adults and adults? Maybe a total length associated with it between females and males would be useful.

Discussion
Needs further work.
Line 157 – et al should not be used on a pers, comm. As we cannot associate to an author if no surnames.
Line 158-161 having a bit of difficulty in following what you mean to say. “This sequence scenario, also a lot of hypothesis in the text and sometimes too many assumptions made when in reality sample size is rather low. Strong comment “corroborate the hypothesis?”. Would recommend the usage of the word suggest e.g. the results may suggest the existence of an ancient occurrence… though this would not be suitable at the start, this would be like a main point of discussion.
Start from the discussion of the results of the study itself and compare with others, then generalise what this could mean.
Line 169 “where the D. nidarosiensis was found, the bottow shows….” Again should be more formal first “where D. nidarosiensis were found, the seabed present similar characteristics, e.g. canyonys and slopes with high depths, suggesting such features would be favourable to this species”.
Line 181 – may be useful have the common name before as to familiarise those with less benthic knowledge.
Line 186-205 mixed results with discussion
Line 216-223 – Would this be suitable to the discussion?
Conclusion
Again stronger conclusion needed, a lot of hypothesis put on discussion. A bit confused with the discussion and conclusion as they seem rather mixed like sentence in line 239-241 that would be a discussion not a conclusion.
Last paragraph repetition of the end of the discussion.
Suggest revision of the titles for each figure, not summarised and Copyright should be put on the photo itself with the year.
Figure 1 – meaning of all the parameters should be added to the legend DL, L1P, etc
Figure 2 – for example “Spatial distribution of Norwegian skate Dipturus nidarosiensis caught during 2010-2016”. Number corresponds to specimens code reported in Table 1.
Figure 4 “Detail of the mouth of (A) female with molar-like teeth and, (B) male with pointed teeth,
Figure 5 “Detail of the snout (A) dorsal and (B) ventral side.
Figure 6 change color for colour, North-Eastern Atlantic Ocean for North-East Atlantic. Add verb “is shown in Table S1”
Table 1 maybe something on the lines of “Summary of data collection: biological parameters (sex, maturity, morphometric parameters, weight and meristic parameters) and source information (commercial or survey, date, depth)”
Commercial should be used instead of landing
Suggest maybe add sex and maturity at a higher level, too far down the line if above then easier to compare females and males and different stage at maturity.
Suggest reducing the number of references. Suggest reading Griffits et al 2011 “levels of connectivity between longnose skate in the Mediterranean sea and the North-Eastern Atlantic Ocean” Conservation Genetics. May be of use considering that these two species may have been confounded in terms of identification.
It should be noted that these above are mere examples as major work is required.

---

## Round 0.2 · Major Revisions

While I recognize that you have produced a new version introducing major changes, I also agree with reviewer 3 when it considers that major revision is still required. Again, I request that you consider all comments made by the reviewers, especially reviewer 3.

I expect to receive a much better version of the article otherwise I will be obliged to not accept it for publication.

Reviewer 1 ·

Basic reporting

The MS was successfully improved, especially the Introduction and discussion sections.

Some comments:

Lines 138-139: MEGA is not a sequence editing software. Please modify this.
Line 213: change “diffusion” by “dispersion”
Lines 214-215: add in final sentence: , but more samples and analyses are needed to confirm this
Line 273: R. radata must be R. radiata, but now this species is named as Amblyraja radiata

Table 1: change “Medits” by “MEDITS”

Experimental design

Experimental design is well supported, and the number of COI sequences here analyzed was clarified.

Validity of the findings

The results bring important new insights on the distribution of D. nidarosiensis in the Mediterranean.

Additional comments

I have now carefully read the revision of this manuscript. The authors took into account the reviewers comments and. I find the manuscript clearer.

Reviewer 3 ·

Basic reporting

Although the subject of this manuscript is interesting, the changes made are not yet enough to make this work published in its current form.
The autors took time to change and improve the manuscript but confusing at times. And although some comments are made, exhaustive details may not have been provided as fear that manuscript could use a reform.
The abstract itself is poorly written and does not capture the main facts needed in an abstract.
Throughout the manuscript terms are not consistently being used, for example landings, discard surveys. I am not sure authors should call offshore observer programmes of "discard surveys", and landings should be clear that we are talking about commercial landings. Also when it comes to research surveys, it should be clear they are scientific surveys, and at some point it is used fishery-independent surveys, need to use one term to define things not several for same things. And these inconsistencies start already in the abstract.
Line 24-25 - "during several biological samplings" not sure author should say several biological samplings, should be more formal, for example something on the lines of: "Biological sampling was conducted on specimens collected through commercial landings, offshore observer programme and scientific surveys." I would not add MEDITS trawl surveys on the abstract (no need for acronyms straight away). Also abstract misleading in terms of the samples, they were already studied in another manuscript so it should also be made clear there that this is the case.
The sentence about the lengths and weights and depths did not change from before even after being suggested by reviewer. And improvements should be made.
There is no need to say in the abstract were the fish were caught... this type of details can be found in the methods.
Strongly dislike the sentence "only partially explored or completely unexplored by fishery and/or researcher vessels" and this sentence occurs more than just in the abstract it should change and it is highly misleading. So is it partially explored by whom, is completely unexplored by whom? and also it should research vessels and not researcher vessels....
Line 37 "has been unnoticed" should be changed to "may have been"

Also inconsistency on the present and past tense use... sometimes verbs are put in the present and sometimes in the past. not very clear.

Introduction - Author should start with the main point of this manuscript... starting with species discrimination is just why? No need to start talking genetics as such.
Would suggest starting with species misidentification is a common issue among several different species, including the species we are interested in, then go to the possiblities of the genetics bring into helping uncovering these misidentifications.
Retract any personal comment on the introduction (here is time for facts not possibilities, these could be mentioned in a discussion section but never on an introduction), line 80-81.
line 86-87 - misleading, these issues are common to all elasmobranchs, so it does not make skates special in that way.
line 89-09 remove the on "the specimens".
line 96-97 elucidating / improving... stronger sentence needed elucidating is a bit over the top. removing also the collected data, I would say this study could futher inform on this species geographical distribution and life-history traits.
The examples here mentioned for the introduction are mere examples, as the introduction needs revamp.
Methods
Sampling starts again with the inconsistencies on what is an offshore programme, and there is no e.g. on here, we are talking about 8 specimens so either they were collected on a commercial vessel, or from the market, or from the survey.
And landings / sicentific surveys / market/ discards cannot all be called survey.
On the sampling it should be clear that some of these already have been studied before, is highly misleading on saying these are brand new methods when they are not.
Lines 112 to 115. are we using term of rays or skates, if needed then should be skates and rays, part of the introduction talks about skates no rays come here, misleading.
Gonadal maturity - it is used on the surveys but has been agreed by whom? for example is it an working group (e.g. under ICES, FAO, etc) protocol, or just related to this.. should be clear where it comes from.
Genetic analyses - line 123 - 124
should change to something on the lines of "Eight specimens were used for the genetic analysis, with four previoulsy analysed by xxx (code 1-4, table 1) and this study new material (code 5-8).
line 126-127 - Total genomic DNA was extracted from tissues samples using xxx, these samples (skin, white muscle) were collected from the ventral side of the pectoral fin (ca. 0.5 cm3 in size).
Results - poor section
statistics? which statistics, we are talking here about raw values, not treated numbers that suffered some statistical analysis so these should be rephrased...
Line 176-181 what do we mean with disc was wider than long? "typically for this species" this is already giving a "discussion" point, and it should not at all. We are merely describing facts in the results section.
"with a number of rows" how many misleading...
remove "included" before between 42-45
182-184 why again starting describing the adults, start from jhuveniles and go from there onwards...
187-188 - repetetion - this should come in the methods in terms of previously analysed not again on the results.
190 suggest "the combination of these data with those available .... Ocean, provided with a final alignment of ,,,,
Past and present mixed line 192 and 193 , was obtained, shows...
Sentence starting in line 192 should stop in South Africa. next sentence to start e.g. " It also showed six additional haplotypes, with four exclusive to xxx and two exclusive to xxx."
Discussion
Line 214 and 215 - how can we say "ancient" or "recent entry" not clear the distintion anywhere in the manuscript.
again 228 -229 needs to be changed, poorly constructed.
Sentences in the discussion starting with Indeed should be revisited, Please do not use "Indeed" that often and it the beggining of a sentence, it seems author is talking to reader instead of describing in written what they think it means.
line 241 - sub-adults should come before adults.
Sentence 243-246 - strong hypothesis .... too low number to suggest anything.... could still be mentioned but with a different approach like for example capture of juveniles only occured on the upper part of the slope, which may suggest juveniles may live in shallower water than sub-adutls and adults...
line 242 - how can you say is a stable local population?
Line 249-253 - these are results there is no evidence of any discussion point on these sentences... and to be honest what are the main difference? juveniles and adults are both in bathyal muds?? it may be that the area where you sampled the adults also a damaged area so you wouldn't have the sea pen in those areas... is highly misleading.
but again these are not discussion points they are results.
line 255 "appears to confirm" should be changed to something like "may support"
line 257-258 again repeat
I would suggest the author to start the discussion with points regarding the morphological parameters rather than genetics, as these are more simple and obvious to describe, with then genetics supporting the first impressions on the specimens.
line 268 - 269 - again a result not discussion point
line 275 - the thorns on the side of the tail in adults are commonly damaged (e.g. through age and damaged by other animals, but also on the capture event) so it is a bit unclear about the condition of the any of the specimens which means again unclear...
line 296-298 again repetition that we need more animals... 2 or 3 times on a discussion no need for this... you can circumnvent by using further data are needed.
line 311 - again start with juveniles
figure 1 - change positions to locations
figure 2 - rapresented change to represented
figure 3 - start with juvenile
figure 4 - remove "the" in "shown in the figure 4"
Table 1 - change title of the table to something like "summary of data collected from commercial landings and scientific surveys, NA: not available)
Table 1 "change survey" to "source", year on date to 2010 not '10, Landing and Medits, so which landings are from market sampling or from the observer programme (discards), if the distinction is given in the methods it should be clear from where these fish come from.
A footnote should be added to the fish which have previously been study in another manuscript. or it should be added in the table caption/title.
Table 2
Gonads - rearrange table to female and male side by side as so you have one column from left to right: stage, maturation state, gonad, with female and male side by side. now in rows.
also inconsistentcy oocytes and eggs, formal term oocytes.
stage 2 how small is a nidamental gland supposed to be? oviducts? uterus? plural and singular...
has this key been used by ICES, FAO or is just MEDITS surveys? has it been approved and scientific reviewed by experts
stage 2 male -"eventually????" not eventually,
yellow eggs? oocytes
any idea about size like < 5mm > 10 mm a bit more reference than just bit white, bit yellow...
sage 3b now we mention oocytes of different sizes???
stage 3 a male their internal structure are not is, we are talking about both claspers...
stage 4a reduced nidamental gland diameter
the whole key needs to be review in terms of english as it is poorly constructed with sentence instead of summary the facts. for example: stage 2 males - "clasper larger than stage 1 and skeleton flexible. Clasper extending past the posterior edge of the pelvic fins (i would imagine you are saying it is that, because if not they are still immature fish). Spermducts well developed, and starting to meander.

Think authors should see what is in terms of ICES WGEF matury key for skates and rays.
Matury keys have been agreed in 2012 workshop.

Experimental design

Ethical compliance - allows misinterpretation starting with all, misleads reader to all have been form fishery.
plus would say from "commercial fishing activity" so it needs rephrasing, like "All .... analysed in this study from commercial fishing activity (DCF xx), do not fit under....
However, specimens brough onboard alive during scientific survey (remove "the" were given an anaesthetic..... either say fishery-independent surveys or scientific surveys, choose one term.
On another note, there is no mention on the whole manuscript to which type of anaesthetic was given to the specimens and that this will not affect any of the results (and if there is, then it should be also mentioned), we should know which type of product was given to each specimen.
And what was the faith of the fish caught alive on the scientific surveys as with anaesthics we are talking about an decrease on their survivability... which I doubt it was as maturity was established which means fish were killed, therefore it was not an anaesthic but in reality killing the fish for the purpose of this study.

Validity of the findings

See above text.

Additional comments

See above comments

---

## Round 0.3 · Minor Revisions

I acknowledge that you had made important changes taking into account all comments made by reviewers. The new version is much better. However, please find attach a PDF with some minor comments that you must take into consideration before sending a final version. Please make sure that your final version is reviewed by an English expert.

---

## Round 0.4 · accepted · Accept

Thank you for the persistence. I found the MS clearer and objective and to be a nice contribution to the knowledge of the skates in European waters.

#